# Evaluating Structural Details’ Influence on Elastic Wave Propagation for Composite Structures via Ray Tracing

**DOI:** 10.3390/s23167220

**Published:** 2023-08-17

**Authors:** Fernando Sánchez Iglesias, Antonio Fernández López

**Affiliations:** ETSI Aeronáutica y del Espacio, Technical University of Madrid, 28040 Madrid, Spain; antonio.fernandez.lopez@upm.es

**Keywords:** SHM, wave propagation, composites, ray tracing

## Abstract

This study presents a novel method based on ray tracing for analyzing wave propagation in composites specifically tailored for structural health monitoring applications. This method offers distinct advantages over the commonly used finite element method mainly in computational resource utilization, which has become a limiting factor for these kinds of analyses. The ray tracing method is evaluated against a number of example cases representing structural details such as thickness changes, stringers, or simulated damage, and the significance of ray tracing to study wave propagation under these conditions and how it can serve as a valuable tool for structural health monitoring are highlighted. This model has been developed as part of a complete SHM framework with the intention of being an efficient and simple way to calculate wave propagation and therefore it could be used as a way to determine relevant damage indicators or train an artificial intelligence model.

## 1. Introduction

Lightweight composite materials have revolutionized various industries by offering superior strength-to-weight ratios, improved mechanical properties, and enhanced design flexibility. Thanks to improvements in the structural analysis discipline, weight reduction in diverse applications such as automobiles or aerospace is achieved either by using less material or by substituting a material with a lighter one that can provide more functionality per unit of weight.

The use of lightweight composites, such as carbon-fiber-reinforced polymers, has gained significant popularity in the aerospace industry. Although they present several advantages related to weight saving and manufacturing costs, these structures may require some special attention as they could present a much wider range of failure modes, such as delaminations, fiber breakage, matrix cracking, and impact-induced damage [1,2,3,4]. For this reason, new and affordable systems must be developed to be able to study the performance of these structures in service. In particular, some of the failure modes can be difficult to detect visually but may cause a significant degradation in the strength of the material and, if not assessed fast enough, could lead to a structural failure of the component.

This is where SHM plays a vital role; by employing various sensing technologies, data acquisition systems, and signal processing techniques, SHM enables real-time or periodic monitoring of composite structures. It allows for the detection, localization, and quantification of damage, as well as the assessment of its severity and progression over time. The system is based on an array of sensors permanently installed in the structure. The extraction of damage-sensitive features from the sensor measurements and the analysis of these features can be used to determine the current state of the structure integrity. Additionally, the system could be used to capture events that could possibly damage the structure and assess them in real time [5], and is even able to consider the effect of corrosion [6] and other environmental factors. These SHM systems could present many cost advantages when compared with traditional inspection methods, and could also be used to improve the life prediction of a structure based on its usage [7,8].

Although there are many kinds of sensors currently available in the industry, ultrasonic inspections have historically proven to be a reliable and cheap way of evaluating manufacturing defects or in-service damage. The main advantage of this method is that, in lightweight structures, elastic waves are able to travel large distances with very low attenuation and are very sensitive to the typical damage of interest [9]. Therefore, a significantly complex structure could be inspected with a very small number of sensors [7,10].

Pb(ZrTi)O${}_{3}$ (PZT) is a cheap and well-known material that has excellent piezoelectric properties [11]. The material properties may be altered with acceptor dopants (lower valence additives) that can play a hardening role in the piezoelectric properties, or donor dopants, which play a softening role [12], and it is therefore ideal for many applications, including generating and sensing elastic waves. Employing elastic waves with a distributed piezoelectric actuator/sensor network based on the pitch–catch method has proven to be very successful for impact damage detection [13,14] and for crack and corrosion localization [15], although a significant effort must be made to extract relevant damage identification parameters [16] and analyze the minimal features that may be captured by an SHM system [17].

Due to the high cost of physical tests, numerical simulations can be especially helpful to predict elastic wave propagation and may help improve the accuracy of these systems [18]. Computational numerical simulations provide a very effective way to capture and understand the structural behavior in terms of the elastic wave response; general purpose computational codes such as time domain spectral finite element methods [19,20] or explicit finite element methods [18,21,22,23] are commonly used to calculate the wave propagation. Numerical methods, however, could present a series of significant disadvantages, such as

Computational Complexity: Finite element or finite difference methods can be computationally intensive, especially for large and complex structures. Long computation times and high memory requirements may limit their efficiency, particularly for real-time or iterative analyses.Grid or Element Discretization: Numerical methods rely on discretizing the structure into grids or elements, which may result in some loss of accuracy, difficulties in capturing fine details, or aliasing effects at high frequencies. The choice of grid or element size can impact the accuracy and computational cost of the analysis.Material damping of boundary damping: specifically for the explicit finite element method, introducing damping elements, either as dashpots or material damping, significantly reduces the stable time increment, making it virtually impossible to solve problems where this effect is relevant to the solution.

For these reasons, and also to increase the reliability of the simulations, multiple different attempts have been made in the literature to solve these problems without resorting to numerical simulations, either analytically [24,25,26] or using statistics or artificial intelligence [27,28]; some of these methods were able to achieve very good results, but the algorithms studied may be too complex to generalize or may be limited in applicability.

A different, very promising idea to solve these problems could be based on the ray tracing method. The ray tracing method is based on the assumption that the particle motion can be modeled as a number of idealized narrow beams (rays) which advance through the medium in discrete amounts. Tracing the paths of individual rays provides valuable information about wave behavior, including wavefront curvature, mode conversion, reflection, and transmission, as the rays interact with objects present along their path.

The ray-tracing-based approach offers significant advantages over traditional numerical methods in terms of computational efficiency and ease of implementation. It provides a practical tool to analyze wave propagation phenomena and has been widely used in scientific research for many different applications, most notably astronomy [29], optical design [30], ocean acoustics [31,32], and heat transfer [33]. Specifically, it has also been previously used in SHM to solve the guided wave propagation problem [34,35,36,37] and it has proven its effectiveness.

The solution presented in this article also applies the ray tracing method to solve the general guided elastic wave propagation problem for an arbitrary shell structure and it aims to provide additional versatility, such as accounting for multiple sensors or actuators in the same problem and introducing adjustable boundary parameters, in order to accurately match experimental data. The objective is to develop an efficient and accurate numerical model that may later be used as part of a complete SHM framework, whether this is to train an artificial intelligence model or to determine relevant damage indicators.

Additionally, the sensor model that has been developed in this article considers a finite area instead of a single spatial point. This provides a significant advantage when compared with other ray tracing methodologies for the following reasons:There is no need to calculate additional eigen-rays to capture the signal at the sensors [31]. The initial ray propagation is already sufficient.In cases where the signal wavelength is comparable to the sensor dimensions (similar to the study cases presented in this article), the effect is already taken into account in the recovered signal, and therefore no additional computation is needed.

However, the main limitation of this approach is that a sufficiently large number of rays must be used to completely capture all possible relevant propagation paths that may reach the sensor.

The set of test cases presented in this article is focused on validating the ray tracing model and demonstrating its applicability to simulate intact structures and detect possible presence of anomalies; however, no attempts are made to characterize the kind or severity of possible damage to a structure.

## 2. Materials and Methods

### 2.1. Ray Tracing Methodology

In general, elastic waves in solid materials are guided by the boundaries of the media in which they propagate. Waves in infinite metallic plates were among the first guided waves to be analyzed in 1917 by Horace Lamb, and have been extensively studied in the literature since then for their applications in SHM ([38,39]). There are three distinctive propagation modes: Symmetric (S0), Anti-symmetric (A0), and Shear (Sh0). To represent the elastic wave propagation, each propagation mode is modeled with independent rays, and the behavior of the waves in the simulation is considered linear and uncoupled.

The propagation velocities of these modes are dispersive and are calculated by the stiffness transfer matrix method (STMM), described in [40]. This methodology is very convenient for multilayered media, as it condenses the system into four equations, eliminating all other intermediate interfaces. Assuming all composite layers behave as orthotropic media and following Hooke’s law generalized for a two-dimensional shell, the layer stiffness matrix in global coordinates (rotated around the laminate perpendicular axis) *C* expresses the relation between layer strains and stresses as:(1)$$\left\{\begin{array}{c} \sigma_{11}\\ \sigma_{22}\\ \sigma_{33}\\ \sigma_{23}\\ \sigma_{13}\\ \sigma_{12} \end{array}\right\}=\left(\begin{array}{cccccc} C_{11} & C_{12} & C_{13} & 0 & 0 & C_{16}\\ C_{12} & C_{22} & C_{23} & 0 & 0 & C_{26}\\ C_{13} & C_{23} & C_{13} & 0 & 0 & C_{36}\\ 0 & 0 & 0 & C_{44} & C_{45} & 0\\ 0 & 0 & 0 & C_{45} & C_{55} & 0\\ C_{16} & C_{26} & C_{36} & 0 & 0 & C_{66} \end{array}\right)\left\{\begin{array}{c} \epsilon_{1}\\ \epsilon_{2}\\ \epsilon_{3}\\ \gamma_{23}\\ \gamma_{13}\\ \gamma_{12} \end{array}\right\}$$
where $\sigma_{ij}$ represent the material stress and $\epsilon_{i}$ and $\gamma_{ij}$ represent the strain and shear angles.

Applying Newtons second law to the previous equation, considering the small linear strain displacement approximation (ϵi,j=0.5∂uj∂xi+∂ui∂xj), the internal forces can be converted into internal stresses, resulting in $\nabla \dot{\sigma }=\rho \ddot{u}$. It is possible to then solve the resulting linear system of equations by imposing a wave solution in displacements of the form $\left(u_{1},u_{2},u_{3}\right)=\left(U_{1},U_{2},U_{3}\right)e^{ik\left(x_{1}+\alpha x_{3}-v_{p}t\right)}$, where $U_{1}$, $U_{2}$, and $U_{3}$ are the amplitudes of the harmonic motions in time and space and $v_{p}$ is the phase velocity of the wave. It is then possible to write the equations of motion as:(2)$$\left(\begin{array}{ccc} C_{11}-\rho v_{p}^{2}+C_{55}\alpha^{2} & C_{16}+C_{45}\alpha^{2} & \left(C_{13}+C_{55}\right)\alpha \\ C_{16}+C_{45}\alpha^{2} & C_{66}-\rho v_{p}^{2}+C_{44}\alpha^{2} & \left(C_{36}+C_{45}\right)\alpha \\ \left(C_{13}+C_{55}\right)\alpha & \left(C_{36}+C_{45}\right)\alpha & C_{55}-\rho v_{p}^{2}+C_{33}\alpha^{2} \end{array}\right)\left\{\begin{array}{c} U_{1}\\ U_{2}\\ U_{3} \end{array}\right\}=0$$

The system in Equation (2) only presents a non-trivial solution when its determinant has a null value; therefore, it can be treated as an eigenvalue problem to obtain the values of α, resulting in a six-order equation with only even coefficients [40,41]. Therefore,
(3)$$\alpha_{1}=-\alpha_{2},\;\alpha_{3}=-\alpha_{4},\;\alpha_{5}=-\alpha_{6}$$

It is then possible to rewrite the expressions for displacements and stresses as follows:(4)$$\left(u_{1},u_{2},u_{3}\right)=\sum_{j=1}^{6}\left(1,V_{j},W_{j}\right)U_{1}e^{ik\left(x_{i}+\alpha_{j}x_{3}-v_{p}t\right)}$$
(5)$$\left(\sigma_{33},\sigma_{13},\sigma_{23}\right)=\sum_{j=1}^{6}\left(D_{1j},D_{2j},D_{3j}\right)U_{1}e^{ik\left(x_{i}+\alpha_{j}x_{3}-v_{p}t\right)}$$

Values for the parameters $V_{j}$, $W_{j}$, and $D_{ij}$ can be found in [40]. Defining a stress displacement vector $s\left(x_{3}\right)=\left(u_{1},u_{2},u_{3},\sigma_{33},\sigma_{13},\sigma_{23}\right)$, the relation for wave propagation in each each layer is:(6)$$\left\{\begin{array}{c} u_{1}\\ u_{2}\\ u_{3}\\ \sigma_{33}\\ \sigma_{13}\\ \sigma_{23} \end{array}\right\}=\left(\begin{array}{cccccc} 1 & 1 & 1 & 1 & 1 & 1\\ V_{1} & V_{2} & V_{3} & V_{4} & V_{5} & V_{6}\\ W_{1} & W_{2} & W_{3} & W_{4} & W_{5} & W_{6}\\ D_{11} & D_{12} & D_{13} & D_{14} & D_{15} & D_{16}\\ D_{21} & D_{22} & D_{23} & D_{24} & D_{25} & D_{26}\\ D_{31} & D_{32} & D_{33} & D_{34} & D_{35} & D_{36} \end{array}\right)\left\{\begin{array}{c} U_{11}e^{k\alpha_{1}x_{3}}\\ U_{12}e^{k\alpha_{2}x_{3}}\\ U_{13}e^{k\alpha_{3}x_{3}}\\ U_{14}e^{k\alpha_{4}x_{3}}\\ U_{15}e^{k\alpha_{5}x_{3}}\\ U_{16}e^{k\alpha_{6}x_{3}} \end{array}\right\}e^{ik\left(x_{1}-v_{p}t\right)}$$

Relating this equation at the top and bottom of each layer, and guaranteeing continuity, it is possible to obtain a relation between the upper and lower surfaces of the plate. Therefore, for a laminate of *n* layers, the expression would be:(7)$$\left\{s_{top}\right\}=\left[T^{n}\right]\left[T^{n-1}\right]...\left[T^{1}\right]\left\{s_{bottom}\right\}=\prod_{j=1}^{n}\left[T^{j}\right]\left\{s_{bottom}\right\}=\left[A\right]\left\{s_{bottom}\right\}$$
(8)$$\left\{\begin{array}{c} \left\{u_{top}\right\}\\ \left\{\sigma_{top}\right\} \end{array}\right\}=\left(\begin{array}{cc} \left[A_{uu}\right] & \left[A_{u\sigma }\right]\\ \left[A_{u\sigma }\right] & \left[A_{\sigma \sigma }\right] \end{array}\right)\left\{\begin{array}{c} \left\{u_{bottom}\right\}\\ \left\{\sigma_{bottom}\right\} \end{array}\right\}$$

Stress-free boundary conditions are then guaranteed in the top and bottom layers as $\left\{s_{top}\right\}=\left\{s_{bottom}\right\}=0$; hence, the dispersion curves can be computed by calculating the determinant $|A_{u\sigma }|=0$

The group velocity, defined as $v_{g}=\frac{dv_{p}}{dk}$, determines the speed at which the wave packet travels, the frequency represents the central frequency of the packet (maximum of the Fourier transform), while the phase velocity, $v_{p}$, determines the speed at which the ray travels, and the dispersion relation can be obtained via these two values.

The shear wave propagation mode (Sh0) is not accounted for in the model, as its usefulness for SHM applications is negligible compared with the other modes and only the S0 and A0 modes are considered and modeled with independent rays.

The ray path along the structure is governed by Snell’s law, and can be solved incrementally assuming linear propagation. The properties of the media are considered uniform until reaching a boundary, so rays can travel on a straight line at each time increment; therefore, the increment time is only limited by the number and density of boundaries in the simulated component and Snell’s Law is only needed to be evaluated at the boundary of two mediums 1 and 2 as:(9)$$\frac{\cos\theta_{1}}{\cos\theta_{2}}=\frac{v_{p1}}{v_{p2}}$$
where $\theta_{i}$ and $v_{pi}$ are the incidence angle and phase velocity of the medium *i*.

### 2.2. Boundary Reflection and Transmission

When a ray encounters a boundary, a component of the ray energy is reflected and another part is transmitted; the model takes into account the mode conversion phenomena [42,43] at each boundary and therefore the transmitted and reflected component energies are divided between the different propagation modes, as shown in Figure 1.

The energy of the incident ray, $E_{iray}$, is divided between the four rays originating at the intersection point and a remnant energy, $\epsilon_{E}$, that is lost at the boundary as follows:(10)$$E_{iray}=f_{1}E_{S0_{r}}+f_{2}E_{A0_{r}}+f_{3}E_{S0_{t}}+f_{4}E_{A0_{t}}+\epsilon_{E}$$
where $E_{S0_{r}}$ and $E_{A0_{r}}$ represent the energy of the symmetric and antisymmetric reflected rays, $E_{S0_{t}}$ and $E_{A0_{t}}$ represent the energy of the symmetric and anti-symmetric transmitted rays, and the factors $f_{i}$ represent the wave energy that is distributed between each of the propagation modes, considering $f_{i}<1$ and f1+f2+f3+f4<1. Both the factors, $f_{i}$, and the dissipated energy, $\epsilon_{E}$, must be adjusted based on the kind of boundary encountered.

For the particular case in which the ray reaches a segment end (i.e., a corner), the ray is eliminated from the model. This approach is taken as a simplification to reduce computational effort, as the effect is negligible for a sufficiently large number of rays and can be minimized considering initial rays that will not intersect any corners of the model on their first propagation step, for example, by choosing an odd number of initial rays.

### 2.3. Ray Signal Recovery

Typical excitation signals used in active interrogation consist of a tone burst, a 50 V peak to peak sine wave modulated by a Hamming window. These signals are named according to the convention BURSTn, where n represents the width of the window in the number of periods. The frequency of these signals is studied in the range of [100, 500] kHz.

This signal information is carried independently by each ray in the frequency domain via a sufficiently large amount of terms in their Fourier transform, and can then be modified as the ray propagates through the model due to different effects, such as material damping. The latter is considered as a factor over the signal amplitude *A*, following an exponential law: $A=A_{0}\exp-\omega \xi t$, where ξ is the material damping coefficient.

After considering dispersion and material damping effects, the ray signal can be obtained during its propagation by applying the time shift property of the Fourier transform $x\left(t-t_{0}\right)⟷e^{-j\omega t_{0}}X\left(\omega \right)$; therefore, one is able to recover the time domain signal at any point over the ray path.

### 2.4. Piezoelectric Sensor Model

Due to piezoelectric sensors’ low weight and their ability to both generate and measure guided elastic waves, they are ideal for SHM systems. Signals obtained unavoidably contain multiple modes requiring complex signal processing techniques to extract useful information. Moreover, PZT sensors may also reveal certain nonlinear behavior and hysteresis under large strains/voltages or at high temperature. Brittleness, low fatigue life, etc., may be some other concerns limiting application ([44]).

To include the PZT sensors in the simulation, they are modeled as a circle boundary. This boundary does not interact with the incident rays; however, the intersection time points ($t_{i}$ and $t_{e}$) are captured and the ray signal is integrated between the line crossing the sensor area ($s_{e}-s_{i}$), as shown in Figure 2. This way allows the sensor to act as an integrator of the material strain over the area it covers ([45,46]), and the solution is obtained by superimposing the integrated signal of all the incident rays.

## 3. Damage Model Evaluation

In order to evaluate the simulation behavior in the presence of damage, the methodology is compared against a physical demonstrator of the front left wing lower cover of the remotely piloted aircraft system LIBIS, designed by the Technical University of Madrid.

The specimen is equipped with an array of eight 12 mm piezoelectric (PZT) sensors, as shown in Figure 3, and the data are recorded with an Acellent SCANGENIE system with a sampling frequency of 48M Hz.

As shown in the figure, artificial damage to emulate the effect of a possible delamination has been introduced in the specimen. Three sets of tests were performed considering the intact structure, the structure with the artificial damage 01, and the structure with both artificial damages. The input signal used consists of a BURST3 at 350 kHz; the test results are presented as an average of three runs.

The simulation was run with 301 initial rays originating from a piezo-electric transducer PZT8. To simplify the analysis, the dispersion curves were calculated via the STMM method for one of the sections and then scaled by the thickness of the other two. Damage areas were simulated as linear boundary segments.

In order to quantify the effect of the damage on the system, both the energy reflected and dissipated in the boundary were adjusted manually based on the shortest linear propagation path, in this case corresponding to between sensors PZT3 and PZT6. The results compared with the tests after the adjustment are shown in Figure 4. Both intact signals and scatter plots are also shown in the figure.

The values adjusted by a single path were then validated with other paths, such as between transducers PZT8 and PZT6. The energy absorption parameters of the damage were adjusted in order to match the amplitude of the first wave package arrival, with this value the results show an acceptable correlation with all sensors studied, as shown in the figure.

The results obtained for the case of artificial damage 1 and 2 are shown in Figure 5. The same parameters adjusted for the artificial damage 1 are applied in damage 2 and the results still show a similar degree of correlation.

From the figures, it can be observed that the results show a very high degree of correlation with the tests; the main differences are due to error accumulation and dispersion calculation in the ray tracing model.

## 4. Stiffened Composite Plates Case Study

To validate the simulations and gain a better understanding of the propagation of elastic waves on aeronautic structures, a testing campaign was designed for two different representative structures: a simple, rectangular composite panel with a quasi-isotropic 7 ply layup and the same panel with a representative T-shape stringer along the center. They were both equipped with an array of eight piezoelectric sensors, as shown in Figure 6, and the data were again recorded with the same DAQ system as in the previous test.

An additional finite element method (FEM) was developed in parallel to act as a benchmark with the ray tracing simulation ([47]). The analysis was performed with Abaqus/Explicit version 2017; the composite solid panel is represented with continuum shell elements (SC8R) and conventional 2D shell elements (S4R) are used for the piezoelectric sensors/actuators. Due to limitations of the FEM code used, the input signal was introduced in the piezoelectric actuator as a temperature variation.

As the simulation must be able to represent the elastic wave behavior as it progresses through the panel, an average element length of 1 mm was used in the model; this ensures that an antisymmetric wave of up to a frequency of 200 kHz can be captured by the model using at least 10 elements per wavelength. The size limitation to model the elastic wave results in a very large mesh of around 600,000 elements and 1,500,000 nodes, as shown in Figure 7.

Both the FEM model simulation and the ray tracing model with 600 initial rays were compared against the tests. The signal results of relevant sensors are shown in Figure 8.

The correlation shows that the ray tracing model can represent the experimental data with a high degree of accuracy, especially when compared with the finite element model, that, although is more costly computationally, presents a similar accuracy for this problem. As with the previous example, the main differences with the experimental data are due to error accumulation and the wave dispersion model.

Although a computationally costly procedure, mapping the ray signals over a square grid can be useful for visualizing the wave propagation over the plate. This is shown in Figure 9, and compared with the FEM model vertical displacement results at *t* = 0.07 ms. The grid used for the ray tracing model has a side length of 2 mm. The energy absorption caused by the stiffener cohesive bond is not represented in the FEM model, whereas in the ray tracing model it is modeled with a boundary loss factor.

## 5. Conclusions

This study shows that the ray tracing method is an efficient way to calculate guided elastic wave propagation for a general 2D geometry with an arbitrarily large number of boundaries or sensors. Compared to the FEM, the new method exhibits several notable advantages: firstly, it allows for a higher level of accuracy in capturing wave phenomena; the method can better represent wavefront curvatures and mode conversions; and it can trivially represent material or boundary damping.

Secondly, the computational efficiency of the proposed method surpasses that of the FEM, making it suitable for real-time SHM applications. The spectral basis functions enable a more concise representation of the wave field, reducing the computational cost while maintaining a high accuracy. This capability facilitates rapid data processing and analysis, enhancing the timeliness of damage detection and localization in composites.

Additionally, the proposed method consists of computing the propagation map and extracting the signal of relevant sensors by superimposing all the intersecting rays in the sensor’s area, as the signal is only evaluated in the sensors, reducing the amount of data and computational time required.

As an example, the FEM simulation of the flat composite panel described in this article took approximately 5 h to run on 32 cores in a high performance computing cluster, while the 800 ray model took approximately 3 min to process the results for each sensor.

The main limitation of this method is similar to the ones presented with other numerical methods such as FEM or SFEM: larger areas require significantly more computational time. Although the ray propagation model has proven to be more efficient for the structures studied, more testing may be required to properly characterize this limitation.

Currently, only isotropic materials or layered othotropic material models have been studied. These elastic wave propagation models are derived originally from Lamb wave theory, which assumes plane stress conditions and therefore is limited to 2D structures.

Nonetheless, the implementation shown is able to solve moderately large-sized problems on a standard laptop without the need for high performance computing resources, which is a significant improvement when compared with other numerical approaches such as the finite element method, which could require up to days of computation on specialized equipment.

The presented test cases prove the applicability of the model in representing characteristic airframe structures and the detection of anomalies; however, no attempt is made to characterize the kind or severity of possible damage to a structure, but could be considered in future studies.

The methodology was implemented with scalability in mind, and it is intended to be refined and improved to be used on more complex geometries. Additional benchmarks against both numerical simulations and physical tests could be performed in future research.

## Figures and Tables

**Figure 1 sensors-23-07220-f001:**
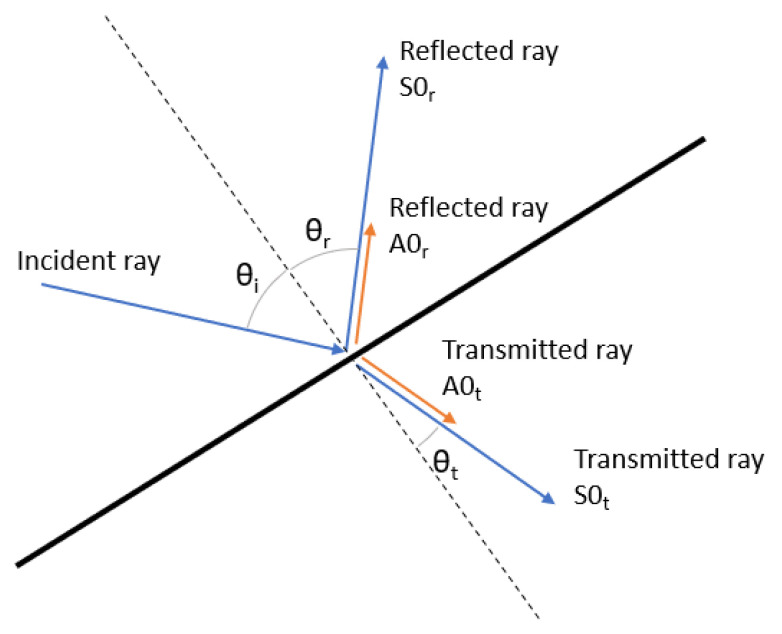
Ray splitting and mode conversion at a boundary.

**Figure 2 sensors-23-07220-f002:**
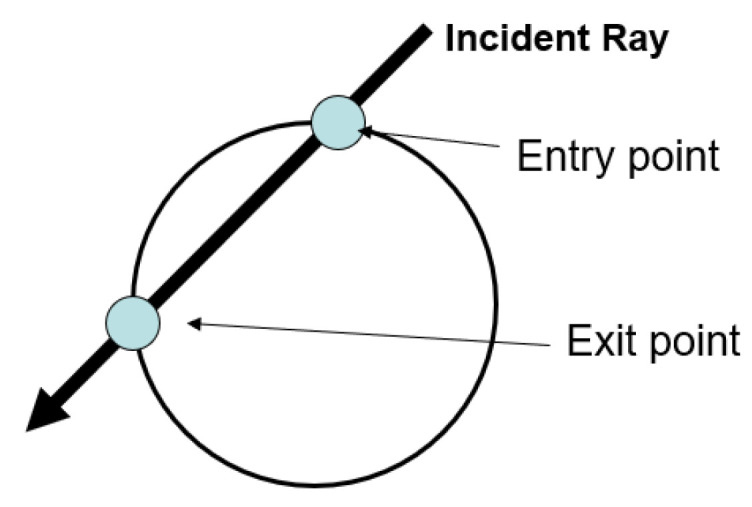
Schema of the PZT sensor model for the ray tracing algorithm.

**Figure 3 sensors-23-07220-f003:**
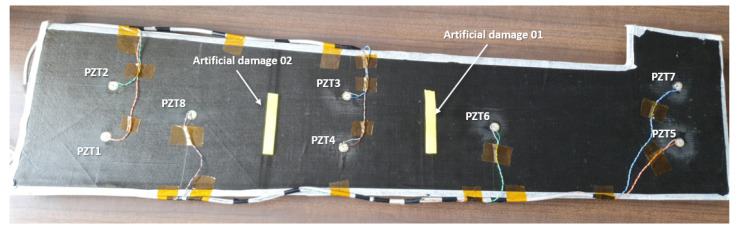
Case of study specimen used for the simulated damage tests.

**Figure 4 sensors-23-07220-f004:**
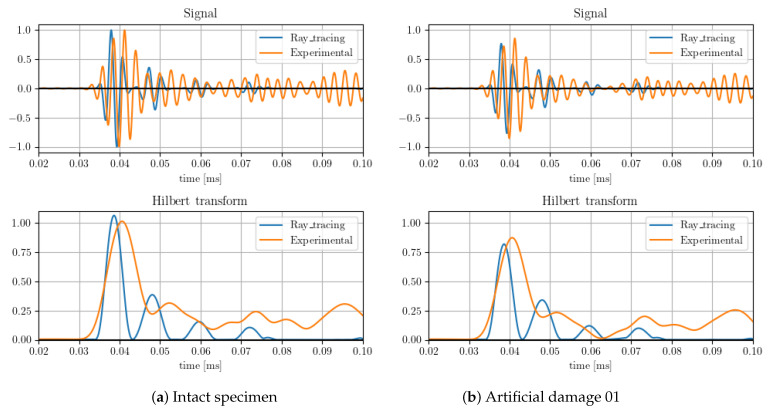
Results comparison of Path 3–6 with artificial damage 01.

**Figure 5 sensors-23-07220-f005:**
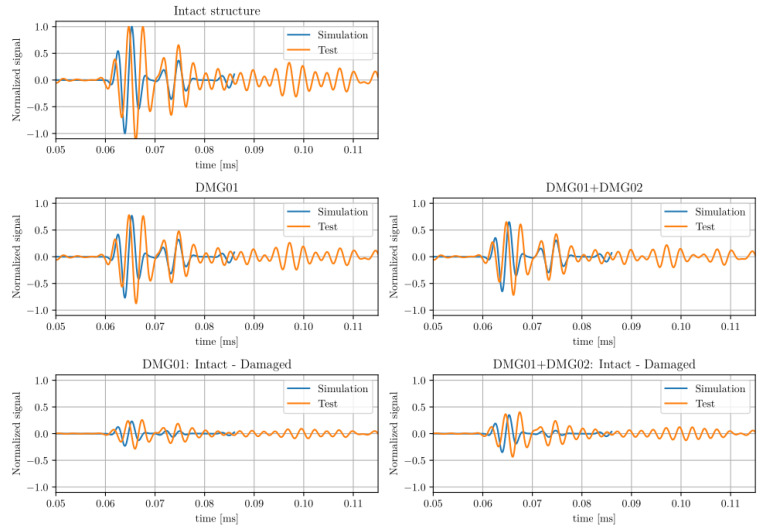
Results comparison of Path 8–6 with artificial damage 01 and 02.

**Figure 6 sensors-23-07220-f006:**
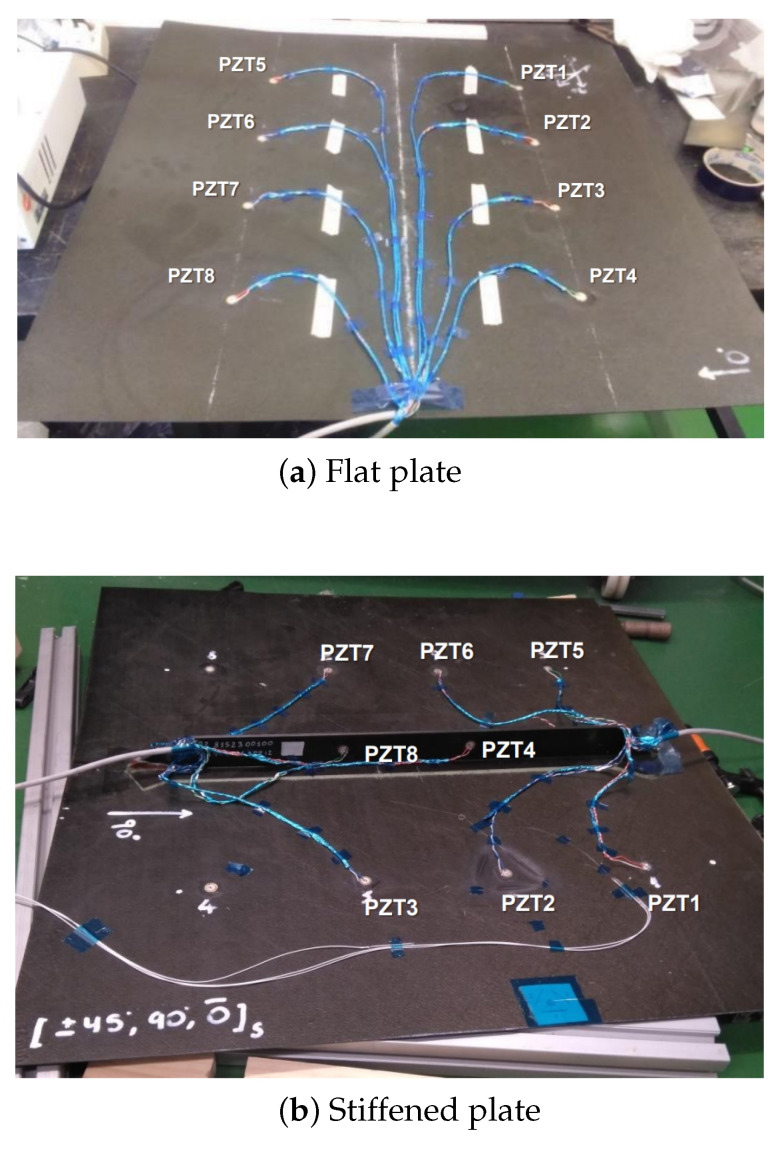
Test specimens for the composite stiffened plate case study.

**Figure 7 sensors-23-07220-f007:**
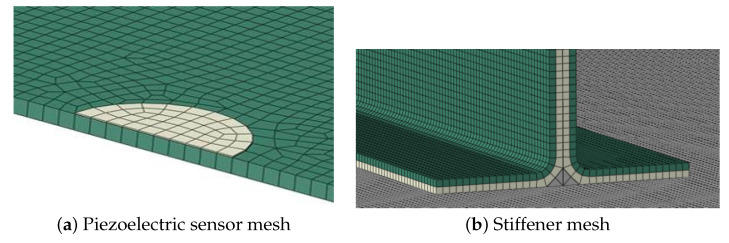
Detail of the FEM model mesh.

**Figure 8 sensors-23-07220-f008:**
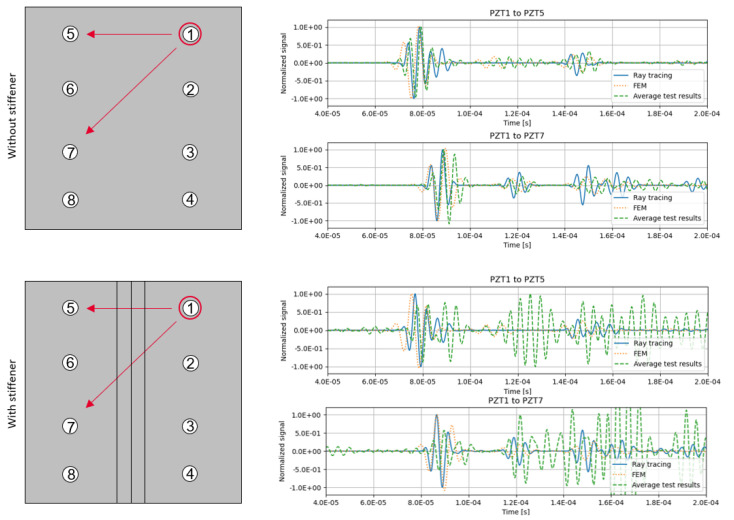
Results comparison of relevant paths from PZT1 for the composite panels.

**Figure 9 sensors-23-07220-f009:**
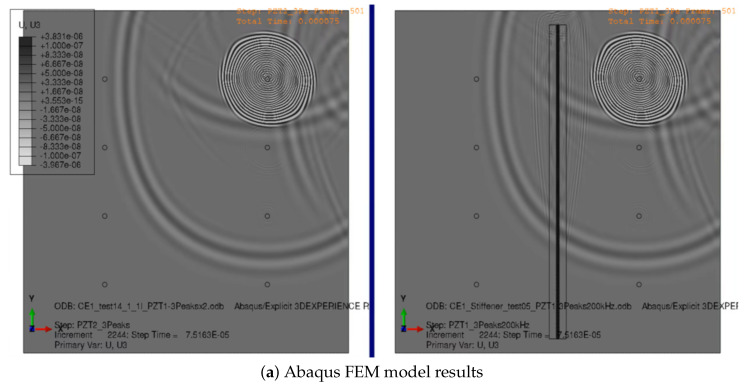
Contour plot comparison at *t* = 0.07 ms.

## Data Availability

Raw data were generated at Technical University of Madrid. Derived data supporting the findings of this study are available from the corresponding author F.S.I. on request.

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
