# Peer review of "Evaluating Structural Details’ Influence on Elastic Wave Propagation for Composite Structures via Ray Tracing"

_sensors, 2023, doi:10.3390/s23167220_

Round 1

Reviewer 1 Report

A ray tracing method was described to calculate the propagation of guided waves in this manuscript. It shows higher computational efficiency than the FEM in applications. However, the following issues should be revised to reconsidered for publication.

1. The execution progress of the ray tracing method should be much more clearly in Section 2. For example, the derivation of formula (1) and (2).

2. What are the reasons induce the difference between the simulation and test signals in Figs. 5 and 8?

3. The limitation of the developed method must be discussed in the manuscript.

4. The parameter definitions are missed in Eqs. (1)~(9).

5. Line 130, “Group velocity, defined as: vg=dw/dk”? It is an error.

Some English writing issues are listed as:

1. Line 40, “…and are are very sensitive….”

2. Lines 89~92, it is difficult to understand.

3. Eq (5), σ23.

4. Lines 168 and 171, the formula expression of exponential law and Fourier transform.

Reviewer 2 Report

Evaluating structural details influence on elastic wave propagation for composite structures via ray tracing was systematically studied by the author. I found the research is intensive and well justified, but with small problems of some English. However, there are still some questions that are supposed to be considered before acceptance for publication

 1.      The authors should re-construct the introduction after they added some more sentences to make the flow of the introduction smoother. The authors are suggested to reconstruct the introduction and put it in a more logical way.

 2.      In introduction, write something about the soft and hard Piezoelectric, and also the specification of different PZTs

 3.      Author should reconstruct the all the figures again.

 4.      There is a font size problem in most of the figures. They should look at the figure again.

 5.      Author should include the recent reference or give a better reason in result and discussion part.

Moderate editing of English language required

Reviewer 3 Report

The paper presents a well-written analysis that explores an alternative approach to structural health monitoring (SHM). However, while the methodology is clear, there is room for improvement in how the simulated and experimental results are compared for damage evaluation. Firstly, it should be noted that the authors primarily focus on anomaly detection rather than damage detection. Therefore, it would be beneficial for the authors to reshape the paper's focus by emphasizing this aspect. Additionally, it is crucial to include appropriate metrics for comparing the experimental and numerical results, such as peak amplitudes, root mean square (RMS), and other relevant measures. Furthermore, the correlation between the intensity of damage and these metrics should be clarified.

To develop a robust damage-detection framework, it is recommended that the authors conduct a sensitivity analysis on their model and correlate these results with the experimental outcomes. This will provide valuable insights into the relationship between the proposed indicators and the actual damage. Furthermore, the paper lacks a clear statement of its final goal. If the authors intend to propose an anomaly detection method, it is important to define the threshold for classifying an anomaly. Consideration should be given to uncertainties based on experimental results. On the other hand, if the aim is to develop a damage detection approach, more extensive work is necessary, as mentioned.

Moreover, the reference list provided is incomplete, and it should consider relevant papers that propose damage detection approaches for SHM. The following references would be valuable additions:

Aloisio, A., Di Battista, L., Alaggio, R., & Fragiacomo, M. (2020). Sensitivity analysis of subspace-based damage indicators under changes in ambient excitation covariance, severity, and location of damage. Engineering Structures, 208, 110235.

Mendler, A., Döhler, M., & Ventura, C. E. (2021). A reliability-based approach to determine the minimum detectable damage for statistical damage detection. Mechanical Systems and Signal Processing, 154, 107561.

These papers offer valuable insights into the development of damage detection approaches within the field of SHM.

Reviewer 4 Report

It seems like the authors only verified the methodologies that already present, to much review contents. The authors should focus on your work rather than the review work. The research gap and the significance of your work should be clearly clarified in the last paragraph of the introduction section, as well as conclusion and the contribution should be summarized in the abstract and conclusion parts.

It is quite uncommon that the affiliation is ONLY the name of the university without the school as well as the country of the affiliation.

It is necessary introduce the background as well as the major conclusion of the study in brief in the abstract.

Too many long sentence and sometimes hard to follow.

Grammer error: Double space in Line 2

In the “Introduction” section, the literature review on related research should be more comprehensive. Some references related to acoustic emission may be helpful, for example: 10.1016/j.engfailanal.2023.107292

Conclusion: The conclusion should be tightened up and this should briefly state the conclusion and contributions of the current research.

It seems like the authors only verified the methodologies that already present, to much review contents. The authors should focus on your work rather than the review work. The research gap and the significance of your work should be clearly clarified in the last paragraph of the introduction section, as well as conclusion and the contribution should be summarized in the abstract and conclusion parts.

It is quite uncommon that the affiliation is ONLY the name of the university without the school as well as the country of the affiliation.

It is necessary introduce the background as well as the major conclusion of the study in brief in the abstract.

Too many long sentence and sometimes hard to follow.

Grammer error: Double space in Line 2

In the “Introduction” section, the literature review on related research should be more comprehensive. Some references related to acoustic emission may be helpful, for example: 10.1016/j.engfailanal.2023.107292

Conclusion: The conclusion should be tightened up and this should briefly state the conclusion and contributions of the current research.

Round 2

Reviewer 1 Report

The comments have been appropriately revised. 

Author Response

Thank you very much.

Reviewer 3 Report

Accept

Author Response

Thank you very much.

Reviewer 4 Report

NO

NO

Author Response

Thank you very much.